# Genetic Diversity and Evolution of Viruses Infecting *Felis catus*: A Global Perspective

**DOI:** 10.3390/v15061338

**Published:** 2023-06-07

**Authors:** Shi-Jia Le, Gen-Yang Xin, Wei-Chen Wu, Mang Shi

**Affiliations:** State Key Laboratory for Biocontrol, School of Medicine, Shenzhen Campus of Sun Yat-sen University, Sun Yat-sen University, Shenzhen 518107, China; leshij@mail2.sysu.edu.cn (S.-J.L.); xingy29@mail2.sysu.edu.cn (G.-Y.X.); wuweixiongde@126.com (W.-C.W.)

**Keywords:** cats, virus, diversity, recombination, evolution, molecular epidemiology

## Abstract

Cats harbor many important viral pathogens, and the knowledge of their diversity has been greatly expanded thanks to increasingly popular molecular sequencing techniques. While the diversity is mostly described in numerous regionally defined studies, there lacks a global overview of the diversity for the majority of cat viruses, and therefore our understanding of the evolution and epidemiology of these viruses was generally inadequate. In this study, we analyzed 12,377 genetic sequences from 25 cat virus species and conducted comprehensive phylodynamic analyses. It revealed, for the first time, the global diversity for all cat viruses known to date, taking into account highly virulent strains and vaccine strains. From there, we further characterized and compared the geographic expansion patterns, temporal dynamics and recombination frequencies of these viruses. While respiratory pathogens such as feline calicivirus showed some degree of geographical panmixes, the other viral species are more geographically defined. Furthermore, recombination rates were much higher in feline parvovirus, feline coronavirus, feline calicivirus and feline foamy virus than the other feline virus species. Collectively, our findings deepen the understanding of the evolutionary and epidemiological features of cat viruses, which in turn provide important insight into the prevention and control of cat pathogens.

## 1. Introduction

Cats (species *Felis catus*, genus *Felis* of the family *Felidae*) harbor a wide range of viruses. Since the discovery of the first feline virus—feline parvovirus in 1928 [1]—more than 25 feline virus species have been identified, among which six have been extensively studied because of known disease association and/or worldwide prevalence. These viruses include feline immunodeficiency virus (FIV), feline coronavirus (FCoV), feline leukemia virus (FeLV), feline calicivirus (FCV), feline parvovirus (FPV) and felid alphaherpesvirus 1 (FHV-1). FCV and FHV-1 are important pathogens that cause feline upper respiratory tract disease (URTD) [2,3], FCoV and FPV are initially primary pathogens of the feline gastrointestinal tract disease [4,5], and FIV and FeLV can cause immunodeficiency disease in cats similar to acquired immunodeficiency syndrome (AIDS) in humans [6,7]. Furthermore, FeLV infection has been associated with quite a few unique diseases, such as acute myeloid leukemia (AML), multicentric lymphoma, aplastic anemia and other immunodeficiencies [8,9,10]. In addition to these well-known pathogens, a number of other feline viruses have been identified, but disease associations remained unclear; these include feline astrovirus (FAstV), feline rotavirus (FRV), feline foamy virus (FFV), Torque Teno feline virus (TTFV) and felis catus papillomavirus (FcaPV). Among these, FAstV and FRV are suggested to be part of the gastrointestinal virome of domestic cats [11], FcaPV is suspected to be associated with cutaneous squamous cell carcinomas (SCC) [12], whereas FFV and TTFV have been regarded as non-pathogenic in felines [13,14,15].

In recent years, with the development of PCR-based and next-generation sequencing techniques, a large number of novel viruses have been identified from cats, including feline morbillivirus (FeMV) [16], feline bocaparvovirus (FBoV) [17], felis catus gammaherpesvirus 1 (FcaGHV−1) [18,19], feline kobuvirus (FeKoV) [20], feline paramyxovirus (FePaV) [21], domestic cat hepadnavirus (DCH) [22], feline picornavirus (FePV) [23], feline norovirus (FNoV) [24], feline chaphamaparvovirus (FeChPV) [25], feline picobirnavirus (FePBV) [26], feline bufavirus (FeBuV) [27] and feline cyclovirus (FeCyCV) [28]. Nevertheless, the pathogenicity of these viruses is currently under-studied, although many have been identified in diseased animals. For example, FBoV, FeKoV, FNoV and FeChPV have been identified in cats with gastrointestinal disease [29,30,31,32,33], FcaGHV−1 from cats with immunosuppressive signs and ocular disorders [18,34,35] and FeMV from cats with renal diseases [16]. Nevertheless, confirming the disease association of these viruses requires more data from standard “case-and-control” studies.

While more viruses are being discovered in cats, our knowledge of the geographic range of cat viruses has also expanded, thanks to a huge number of sequence-based molecular epidemiological studies, each targeting a specific pathogen from a specific region. Indeed, for most well-known pathogens, namely, FCV, FIV, FPV, FeLV and FCoV, their geographic range has been shown to range from the place of discovery to worldwide. Global distribution has also been established for recently discovered viruses. For example, FeMV was first discovered in Hong Kong in 2012 [16], and by using PCR primers designed based on the first few genomes, it was then identified in Japan, the United Kingdom, Brazil, Turkey and Italy [36,37,38,39,40,41,42,43]. Similarly, FeChPV was first discovered in 2019 from a feline shelter in Canada [5] before its presence was expanded to Turkey [44], Italy [25] and China [45,46].

In general, recent studies based on molecular sequencing have witnessed a great expansion of virus species richness, intra-specific genomic diversity, and geographic range for feline viruses. While these diversities are mostly described in numerous regionally defined studies, there is a lack of an overview of global diversity for the majority of cat viruses. As a result, the general evolutionary and epidemiological characteristics of these viruses were never systematically compared. In this study, we downloaded the entire collection of genetic sequences for every feline virus identified thus far and conducted a comprehensive phylodynamic analysis. Our results revealed and compared the overall diversity, geographic and temporal dynamics, disease characteristics and recombination frequency of each cat virus, deepened our understanding of evolutionary and epidemiological features, and provided important insight into the prevention and control of cat pathogens.

## 2. Materials and Methods

### 2.1. Collection of Feline Virus Sequences and Information

The taxonomy of feline viruses was determined based on a comprehensive search of the NCBI PubMed database using the keyword “feline virus” or “cat virus” (Appendix A). For each feline virus species, we downloaded all the nucleotide sequences, coding DNA sequences (CDS) and the associated sequence information, including GenBank number, virus strain, collection time, country of origin, PubMed (publication) ID and host information, from the NCBI GenBank database. For those with missing or incomplete information and for virulence, vaccine, and pathogenic information, an in-depth search was carried out to look for corresponding data from the original publications. All sequence and information collections were carried out in February 2022.

### 2.2. Data Processing and Sequence Alignment

For each virus, the CDS were first categorized based on encoded genes. Sequences with <300 bp in length and >20% of ambiguous nucleotides were treated as low-quality data and removed from the data set. Highly identical sequences (>99% nucleotide identity) were also removed by using Cluster Database at High Identity with Tolerance (CD-HIT) (version 4.8.1) and CD-HIT-EST (version 4.8.1) programs [47], given that they were from the same geographic regions. Based on the number of sequences available, sequence divergence level, and the representativeness of diversity and geographic distribution, it is subsequently determined which gene (and regions of genes) was used to represent the overall diversity of this virus. Corresponding sequences were then aligned using the program Mafft (version 7.480) [48] for later phylogenetic and evolutionary analyses.

### 2.3. Phylogenetic Analyses

The intra-specific phylogenetic trees for most of the viruses involved in this study were reconstructed based on gene or partial gene alignments representative of the virus diversity (see Section 2.2) using the maximum likelihood (ML) algorithm, General Time Reversible (GTR) nucleotide substitution model, and the subtree pruning and regrafting (SPR) branch-swapping algorithm implemented in software PhyML (version 3.0) [49]. For some of the newly discovered viruses, such as FAstV, FNoV and FRV, the inter-specific phylogenetic trees were reconstructed to demonstrate their diversity in the context of other related virus species. The inter-specific phylogenetic trees were reconstructed based on RNA dependent RNA polymerase (RdRp) (RNA viruses) and non-structural protein 1 (NS1) (FeChPV) protein alignments and utilizing a maximum likelihood algorithm, the Le Gascuel (LG) amino acid substitution model, and the SPR branch-swapping algorithm implemented in PhyML. All trees generated in this study were mid-point rooted and were labeled and demonstrated using the ggtree (version 3.2.1) [50] software package implemented in R (version 4.1.2) [51].

### 2.4. Analyses of Phylogeographic Structures

To assess whether the spread of a virus was confined by geographic distances or locations, we used the package adegenet (version 2.1.10) [52] implemented in R (version 4.1.2), which reveals the genetic structure present among geographic regions using discriminant analysis of principal components (DAPC). We also used Wilks’ lambda to evaluate the degree of geographic structure with the MANOVA method.

### 2.5. Analyses of Temporal Structure, Evolutionary Rate and Estimating Time to the Most Recent Common Ancestor (tMRCA)

Before time-scale analysis, sequences without sampling date information were removed from the alignment. The temporal structure of each cat virus species was examined using TempEst (v 1.5.3) [53], which carried out a regression of phylogenetic root-to-tip distances against the sampling date. In addition, time-scale trees were inferred using TreeTime (version 0.8.6) [54] and Least-squares dating (LSD) (version 0.3beta) [55], which transforms ML trees topology into time-scale trees with GTR nucleotide substitution model and autocorrelated molecular clock model.

### 2.6. Analyses of Genomic Recombination

To investigate the frequency of recombination for each cat virus species, phylogenies were reconstructed at the beginning (i.e., the first 2000 bp) and the end (i.e., the last 2000 bp) of the full genome alignments. The trees based on the two genomic regions were subsequently compared for (i) incongruences in topology or (ii) inconsistencies in pairwise genetic distance matrices as measures of recombination frequency. The incongruences in topology were visualized using the dendextend package (version 1.15.2) [56] implemented in R (version 4.1.2). The degree of inconsistencies was estimated based on pairwise comparisons of patristic genetic distance matrices and using the mantel test [57].

## 3. Results

### 3.1. Overview of Feline Viruses around the World

We collected a total of 12,377 sequences of feline viruses from NCBI GenBank database, among which the majority are associated with the five most common feline pathogens, namely, FIV (*n* = 3624), FCoV (*n* = 3066), FeLV (*n* = 1571), FCV (*n* = 1441) and FPV (*n* = 929) (Figure 1A), which also had worldwide distributions (Figure 1B). While FIV sequences are quite abundant across different continents, other viruses have a more uneven distribution. For example, FCoV is more frequently sampled in Asia and Europe than in North America, whereas FeLV is more frequently sampled in Asia and the Americas than in Europe (Figure 1B). Nevertheless, some of the unevenness in distribution might reflect sampling bias instead of true prevalence. Indeed, the sampling sizes in Africa and South America are much smaller than those of the other continents (Figure 1B). Additionally, within Europe, more sequences are obtained from Western than Eastern Europe. Collectively, these observations suggest wide distribution and potential gaps in the global sampling of feline viruses.

### 3.2. Phylogenetic Analysis

We performed phylogenetic analyses that revealed the diversity backbone for more than nine major cat virus species. Among these, FIV is an important cat pathogen whose infection usually leads to the depletion of CD4+ T cells and causes AIDS-like diseases in cats [58]. The virus has been previously divided into seven different clades (A–F and U), each associated with specific geographic locations [59,60,61]. We analyzed 831 representative sequences, which cover 501 nucleotides (nt) (501 bp out of 2571 bp complete length) in the *env* gene. The phylogenetic tree reveals a diversity comprised of at least six well-supported clades. The division is mostly consistent with the sub-typing system from previous studies, except for Clade E, which is merged into Clade B due to a lack of clear distinguishment between the two (Figure 2). In comparison with previously defined clades (A–F), the clade depicted here contains a much larger diversity and wider geographic ranges (Figure 2). Among these, Clades A–D harbor most of the sequences (816/831) identified from the field. Clade A and B are both worldwide clades with global distributions, although, within these clades, there is some degree of geographic structuring of virus diversity (Figure 2). In comparison, Clades C and D are more regionally defined: Clade D is only identified in Asia, whereas Clade C is mainly identified in both Oceania and Asia and contains a highly virulent strain, CABCpady00C, sampled from Canada, which causes high death rates from acute phase immunodeficiency disease [62,63] (Figure 2). Two vaccine strains are available for FIV, which belong to Clade A (i.e., strain Petaluma) and D (i.e., strain Shizuoka), and are used as inactivated dual-subtype FIV vaccines.

FCoV can cause two distinct types of diseases: some cause only mild (often subclinical) gastrointestinal illness, while others can lead to fatal multisystemic disease of feline infectious peritonitis (FIP) [64]. However, these diseases are not associated with specific FCoV genotypes or subtypes, and FIP arises as a result of viral mutations occurring following FCoV infection [65]. FCoVs are classified into two serotypes, Type I and Type II FCoVs [66]. We analyzed 107 complete S gene sequences (4494 nt) of FCoV in the context of 147 sequences under Alphacoronavirus 1, which includes FCoV, Canine coronavirus (CCoV), Transmissible gastroenteritis virus of swine (TGEV) and Porcine respiratory coronavirus (PRCoV) (Figure 3). Type I and II alphacoronaviruses are separated by an average of 58% nucleotide sequence divergence. FCoV is identified in both types, with the majority (92/107) of Type I and only a few (15/107) of Type II, and the latter show a relatively close relationship with Type II CCoV compared with TGEV and PRCoV (Figure 3). Highly virulent strains are identified in both Type I and II viruses, which include the HRB-17 strain from China [67], the 79–1146 strain from the USA [68] and the 26 M strain from the UK [69]. A vaccine strain is occasionally used to prevent FCoV infection, and it is derived from strain DF-2, a Type II FCoV.

FeLV infection in cats is associated with various health issues, including anemia, immune system-related diseases and cancer [70], and it has relatively high fatality rate [71]. We analyzed 278 representatives of the partial *env* gene (1756 bp out of 1929 bp complete length), which divided the FeLV diversity into two major clades, namely, Clade 1 and 2 (Figure 4). For each clade, a significant part of the diversity is associated with viruses sampled in Asia, whereas those from the Americas and Europe form geographically defined lineages that are nested within the diversity of Asian lineages. Furthermore, Asian viruses dominate the sequences (187/278). Two vaccine strains (i.e., FeLV Rickard and FeLV Glasgow-1) are identified in the phylogeny, and they all belong to Clade 1.

FCV commonly infects the respiratory tract of cats, and some strains with higher virulence can cause virulent systemic disease (VSD) [72,73,74,75]. A total of 801 representatives of partial viral protein 1 (VP1) genes (420 bp out of 2007 bp complete length) are used for phylogenetic analysis, which identifies seven well-supported clades (Clade 1–7) (Figure 5). Most of the diversity is dominated by sequences identified from Europe, a continent with much higher overall sequence numbers (644/801) and diversity than the rest of the continents. Interestingly, Clades 2, 3, 6 and 7 are all “cosmopolitan” with geographic expansion into more than three different continents. These clades are also ones that contain more strains with higher virulence and transmissibility than the rest of the clades (Figure 5). Specifically, Outbreak NSW 2015 [76], Outbreak ACT 2018 [76] and Outbreak Massachusetts 2001 [77] were from Clade 2, whereas Outbreak QLD 2017 [76], Outbreak Missouri 1995–1996 [77], Outbreak Sacramento 1998 [78], Outbreak Harrisburg 2003 [78] and Outbreak Florida 2003 [78] were from Clade 6 (Appendix A). Four vaccine strains are identified here, which belong to Clade 6 (FCV-F9) and Clade 7 (F4, FCV-255 and FCV-2024).

FPV is known as an important pathogen that causes gastrointestinal disease in cats [79]. This disease is also referred to as feline panleukopenia or feline infectious enteritis, which can result in severe or even fatal disease in kittens. We used 252 representatives of the complete minor capsid protein (VP2) gene (1755 bp) of FPV for phylogenetic analysis. All FPVs form a sister clade to canine parvovirus (CPV). Within FPV, the diversity can be divided, with low confidence, into two major clades and a number of small transmission chains close to a common ancestor of all FPVs (Figure 6). Asian strains dominate the entire phylogeny (144/252), followed by Europe (42/252) and Australia (19/252). Interestingly, FPV also contains a number of strains identified from non-cat feline hosts, such as the fox, lion, dog and raccoons, amongst others, resulting in gastroenteritis- and leukopenia-type diseases similar to those observed in cats [79,80]. Six vaccine strains (FPV-Felocell, FPV-Panocell, Philips Roxane, FPV-Purevax, PLI-IV and FPV-Dohyvac) are identified within Clade 2 (Figure 6). The outbreak strains, which were recorded in Outbreak Mildura 2015, Outbreak Sydney 2016–2017 and Outbreak New Zealand 2016 [81], are also located within Clade 2 (Appendix A).

FHV-1 normally infects the upper respiratory tract of cats and causes viral rhinotracheitis [82]. We collected 66 complete genomes for phylogenetic analysis, and it revealed three major clades with strong geographic structures, although there are a few genome sequences from continents other than North America and Oceania (Figure 7). Clade 1 is comprised of mainly North American strains, Clade 3 is comprised of only Australian strains and Clade 2 is comprised of strains sampled from both continents. Three vaccine strains (i.e., Companion, Merial Purevax MLV and Feligen) were identified, and they all belong to Clade 1.

FFV is generally considered to be non-pathogenic in domestic and wild felids [15]. FFVs are classified into two subtypes, the F17/951-type and FUV-type [83]. We collected 104 *env* gene (2949 bp) sequences of FFV for phylogenetic analysis. FFV strains can be divided into two subtypes, including F17/951-type and FUV-type (Figure 8). Interestingly, the majority of the sequences obtained here were from cougars sampled from the United States [84]. Among the cat-associated FFV, the majority of the viral sequences are obtained from North America and Asia, although the sampling size is too small for reliable phylogeographical inference.

FeMV is believed to be associated with renal disease, although more data from clinical studies are required to confirm this [38,85]. Previous analysis of FeMV genomes divided the diversity into two genotypic lineages, FeMV−1 and FeMV−2 [86]. In our study, a total of 186 sequences of RdRp gene sequences were analyzed, which divided the diversity into at least three clades (Figure 9). Among these, Clade 1 has cosmopolitan distribution except for Australia, Clade 2 is mainly found in Europe, South America and Asia, and Clade 3 had only four sequences which were all from Europe (Figure 9).

FBoV was often identified in the intestinal tract of cats and was suspected to be associated with gastrointestinal signs [29,30]. To date, three FBoV genotypes have been identified, namely, FBoV−1, FBoV−2 and FBoV−3. A total of 114 NS1 sequences were analyzed and compared. Three types of FBoVs have been identified, including FBOV−1 (*n* = 74), FBoV−2 (*n* = 33) and FBoV−3 (*n* = 7) (Figure 10). The majority of the sequences identified here were sampled in Asia (110/114), although strains from North America and Europe have also been identified (4/114).

In addition to the common feline viruses above, we also investigated the diversity and geographical distribution of less common or more recently discovered viruses. Interspecies phylogenetic analyses reveal that several viral families, namely, *Anelloviridae*, *Astroviridae* and *Papillomaviridae*, contain multiple virus species associated with cats, although no obvious disease association has been identified. Indeed, most of the feline viruses were identified from fecal samples, including FAstV [87], FNoV [24], FeCyCV [28], FeKoV [20], FeChPV [31] and FePBV [88]. Other virus species were identified from samples such as from blood (DCH [89] and FcaGHV−1 [19]) and the gut (FeBuV [27]). Furthermore, among these newly identified viruses, FcaGHV−1 has been identified from more than three continents despite their recent discovery, suggesting that latent or subclinical infections exist in a wide range of cat populations worldwide (Figure 11).

### 3.3. Phylogeographic Structure Analysis

Phylogeographic structures were examined for a few feline virus species with relatively large sample sizes, namely, FCV, FIV, FPV, FCoV, FeLV, FHV-1 and FFV. Based on Wilk’s lambda, all seven viruses examined here had significant geographic structure (*p* < 0.001), although the degree of such structuring varied between different virus species (Table 1). Indeed, the differentiation of geographic groups is most obvious for FCoV, FeLV, FHV-1 and FFV but least obvious for FCV (Table 1). Similarly, DAPC scatter plots reveal clear geographic structure for all viruses examined except for FCV, which shows substantial mixing of virus diversity from different continents (Figure 12). In addition to FCV, there are also clear indications within other viral species where strains from different continents are indistinguishable (i.e., the mixing of North and South American strains in FIV and European and Asian strains in FPV) (Figure 12), suggesting cross-continent transmissions of the corresponding pathogens.

### 3.4. Time-Scale Phylogenies and Analysis of Evolutionary Rates

To reveal temporal and epidemiological characteristics of the feline viruses, we first performed root-to-tip regression analysis, which shows that the temporal structures are poor for most of the viral species (Figure 13A), and therefore we only performed time-scaled evolutionary analyses for those with R > 0. Two approaches (i.e., TreeTime and LSD) were used for evolutionary rate and tMRCA estimations, and they generated consistent results for the five viruses examined here (Figure 13B,C and Appendix A). The evolutionary rate significantly varied among different virus species. Among these, FCV and FCoV Type II have the fastest evolutionary rate, at a median of 6.71 × 10^−4^ (CI 4.57 × 10^−4^–7.89 × 10^−4^) and 2.778 × 10^−4^ (CI 9.33 × 10^−5^–6.50 × 10^−4^) substitutions/site/year based on LSD method, and 7.82 × 10^−4^ (CI 6.92 × 10^−4^–8.72 × 10^−4^) and 9.25 × 10^−4^ (CI 8.65 × 10^−4^–9.85 × 10^−4^) substitutions/site/year based on TreeTime method. Despite the fast evolutionary rate, the tMRCA for current FCV diversity was traced back to 1449 (CI 1381–1517) and 1428 (CI 1160–1516) using TreeTime and LSD methods, respectively, suggesting the early emergence of the FCV. Furthermore, the lowest evolutionary rate was estimated for FPV, an ssDNA virus, which has 4.11 × 10^−5^ (CI 2.11 × 10^−5^–6.11 × 10^−5^) substitutions/site/year and 5.29 × 10^−5^ (CI 2.95 × 10^−5^–6.29 × 10^−5^) using TreeTime and LSD approaches, respectively.

### 3.5. Recombination Analysis

We assessed recombination frequencies for different viral species by comparing phylogenies reconstructed based on the first and last 2000 bp of the full-length genomes, which included FCV, FCoV, FFV, FPV, FeLV and FeMV. Correlation coefficiency estimations based on patristic distances (i.e., genetic distance derived from the maximum likelihood phylogenetic trees) revealed relatively greater phylogenetic incongruence for FPV, FCoV, FCV and FFV (Pearson’s coefficient < 0.8) and therefore higher recombination rates for these viruses (Figure 14). Strikingly, the coefficient for FPV was as low as 0.153 (*p* = 0.067), suggesting extremely frequent genomic exchange between diverse FPV genomes. Similarly, low recombination rates were inferred for FeMV and FeLV, which had highly congruent phylogenetic structures between the 5′ and 3′ ends of their genomes (i.e., coefficient > 0.8) (Figure 14).

## 4. Discussion

In this study, we obtained a thorough collection of sequence data associated with viruses infecting cats. We revealed, for each of the viruses, the most comprehensive diversity described to date. Specifically, our data set has greatly expanded the knowledge on the diversity of FCV [75,90,91,92,93], FPV [4,94,95,96], FCoV [69,97,98,99], FIV [60,61,100,101,102] and FeLV [8,103,104,105,106], such that in-depth analyses of feline viruses from across the globe could be performed. From there, we gained important insight into the global distribution, geographic spread pattern, evolutionary time-scale and recombination frequencies of these viruses. Generally, a larger diversity and wider distribution are described for the majority of viral species. Nevertheless, the diversity depicted here was by no means all-inclusive for these viruses. This is because the global sampling was largely uneven, with more diversity revealed for developed countries than for developing ones. Indeed, the number of sequences collected from African countries was substantially smaller than those from Asia (mainly China and Japan), European, and North American countries. Furthermore, viruses with low pathogenicity or less clinical impact, namely, TTFV, FFV, FePV, FePBV, FeCyCV and FePaV, contained fewer sequences and poorer geographical representation, most likely due to lack of surveillance from “targeted sequencing”. The global prevalence of these viruses is expected to be much higher, given that many cause subclinical or latent infections in cats.

Despite the global distribution of feline viruses, before this study, there was a lack of an overview of how they spread across the globe. This study revealed two distinct types of geographic structure: a few viruses or viral lineages showed geographical panmixes that features the dissemination of closely related virus over a long distance, whereas the others are more geographically contained. It is still unclear what causes the rapid transmission of cat viruses across the globe. One possibility is the long-distance transportation of pets [107,108,109,110]. Alternatively, the involvement of contaminated fomite or a secondary host could not be ruled out. Regardless, poor geographic structure and rapid long-distance distribution were mostly observed in FCV, which was a respiratory pathogen with frequent outbreaks of highly transmissible viral strains [76,111,112,113], implying that the mode of viral transmission might be a key contributor to the rapid spread of viral pathogens globally.

Our study revealed evolutionary rate and tMRCA estimations for a few RNA viruses (i.e., FCV, FCoV), retroviruses (i.e., FFV) and ssDNA viruses (i.e., FBoV and FPV). Our estimations of the median evolutionary rate of FCV were 7.82 × 10^−4^, which was much lower than a related virus, human norovirus, at 4.16 × 10^−3^ substitution/site/year [114] but similar to that of rabbit calicivirus at 7.7 × 10^−4^ substitution/site/year [115]. The rate estimation for FCoV and FPV was 9.25 × 10^−4^ and 4.11 × 10^−5^ substitution/site/year, which was similar to previous estimations for FPV [116] and coronavirus (e.g., SARS-CoV-2) [117] at 9.4 × 10^−5^ and 1.69 × 10^−3^ substitution/site/year, respectively. Accordingly, the estimation for tMRCA revealed that the FCV was circulating in the global cat population for at least 750 years, whereas FFV for only 90 years. Nevertheless, caution must be taken in interpreting tMRCA for FFV because the sample size obtained thus far is very limited, and it may not reflect the true diversity of the virus, and therefore the tMRCA might be massively under-estimated. Furthermore, our results also revealed poor temporal structure for a number of viruses, including FIV, FeLV and FHV-1. This is expected because these viruses, all of which are retroviruses or DNA viruses, have a slow evolutionary rate such that their time-scales are unlikely to be inferred from contemporary sampled sequences [117].

Recombination is an important driving force of evolutionary diversity and has been frequently reported for positive sense RNA viruses and DNA viruses [118]. In our study, the highest recombination rate was observed for FPV, which is a ssDNA virus. Although FPV has a low evolutionary rate, its recombination rate is high and is the main driving force of its genetic diversity, as expected for other ssDNA viruses [116]. For RNA viruses, the highest recombination rates were observed in FCoV and FCV, belonging to *Coronaviridae* and *Caliciviridae*, respectively, both of which were among the RNA viruses families with the highest recombination rates [118]. Indeed, the recombination hotspot of FCV is located at the breakpoint between the nonstructural protein coding region (Open reading frame 1(ORF 1)) and the structural protein coding region (ORF 2) [119]. Whereas that of FCoV is also located between the structural (Spike) and nonstructural (ORF1b) proteins [120]. Furthermore, it has also been reported that the Type II FCoV sequence originated from recombination of the Type II CCoV sequence and the Type I FCoV sequence [121], suggesting recombination as a common mechanism that drives the evolutionary diversity of positive sense RNA viruses in general.

## 5. Conclusions

In summary, our study performed an all-inclusive sequence collection from the GenBank and comprehensive phylodynamic analyses for 25 feline virus species. Our results revealed, for the first time, a global diversity much larger than previously depicted, based on which we further characterized the geographic expansion patterns, temporal dynamics and recombination frequencies of these viruses. Importantly, the representative sequence data set obtained here forms a knowledge basis for cat virus diversity, evolution and epidemiology, and they could be used as reference data sets or diversity backbones for future surveillance work.

## Figures and Tables

**Figure 1 viruses-15-01338-f001:**
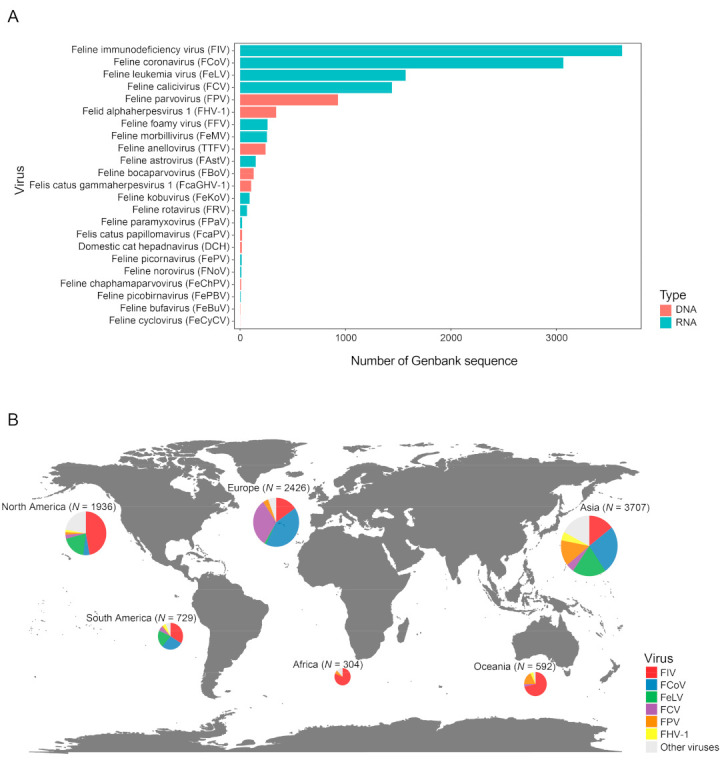
Overview of the distribution of feline virus sequences. (**A**) Number of GenBank sequences available for each feline virus species. Red and blue bars represent DNA and RNA virus, respectively. (**B**) The global distribution of virus sequences collected for this study. The size of the circle reflects the number of sequences.

**Figure 2 viruses-15-01338-f002:**
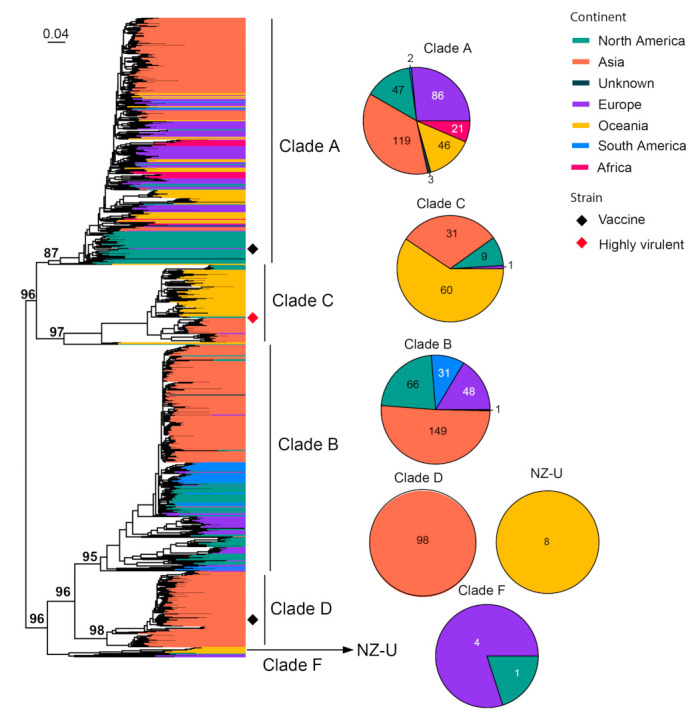
Diversity and phylogeographic structure of FIV based on env gene. A maximum-likelihood tree reconstructed based on partial *env* gene alignments (501 nt) is shown on the left, which includes 831 virus sequences obtained from the GenBank. The tree is mid-point rooted. Geographic locations (continent) for these virus strains are marked with different colors. Highly virulent strains and vaccine strains are labeled using red and black diamonds, respectively. The detailed geographic distributions of each subtype are shown as pie charts on the right of the phylogenetic tree.

**Figure 3 viruses-15-01338-f003:**
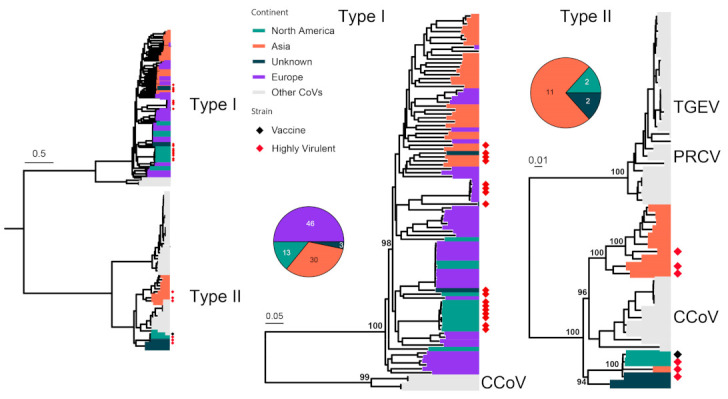
Diversity and phylogeographic structure of FCoV based on S gene. A maximum-likelihood tree reconstructed based on S gene alignments (4494 nt) is shown on the left, which includes 147 virus sequences obtained from the GenBank. The tree is mid-point rooted, and detailed subtrees of each serotype (i.e., Type I and II) are shown on the right. For each serotype, the geographic distributions are shown as a pie chart and placed left next to the corresponding subtree. Geographic locations (continent) for these virus strains are marked with different colors. Highly virulent strains and vaccine strains are labeled using red and black diamonds, respectively.

**Figure 4 viruses-15-01338-f004:**
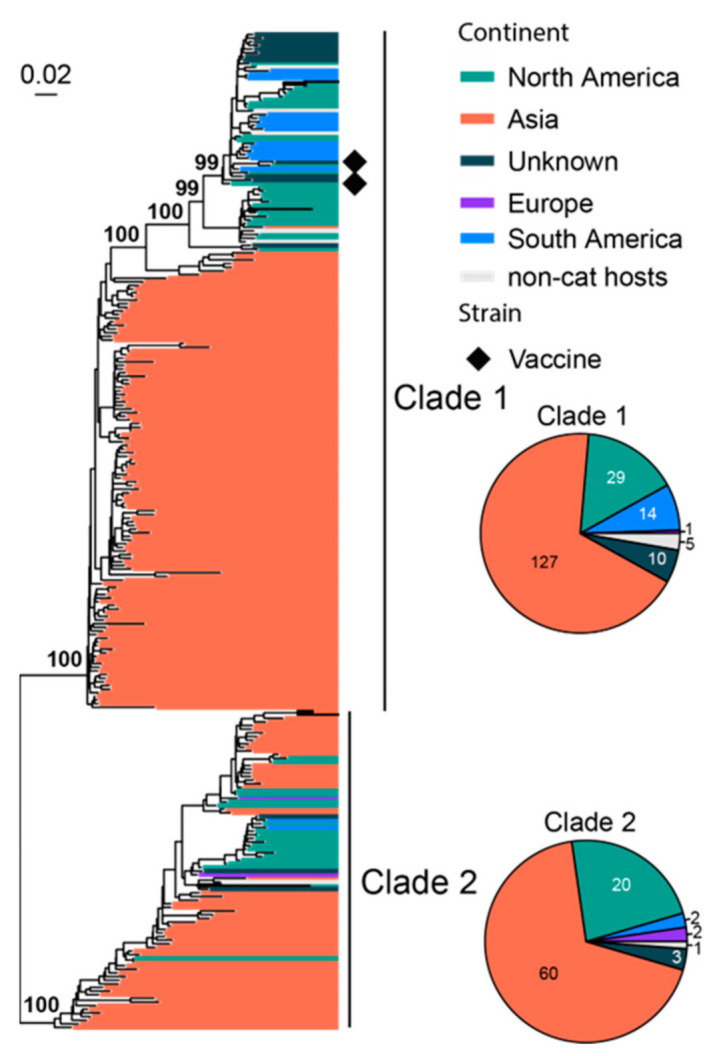
Diversity and phylogeographic structure of FeLV based on partial env gene. A maximum-likelihood tree reconstructed based on partial *env* gene alignments (1756 nt) is shown on the left, which includes 278 virus sequences obtained from the GenBank. The tree is mid-point rooted. Geographic locations (continent) for these virus strains are marked with different colors. Vaccine strains are labeled using black diamonds. The detailed geographic distributions of each subtype are shown as pie charts on the right of the phylogenetic tree.

**Figure 5 viruses-15-01338-f005:**
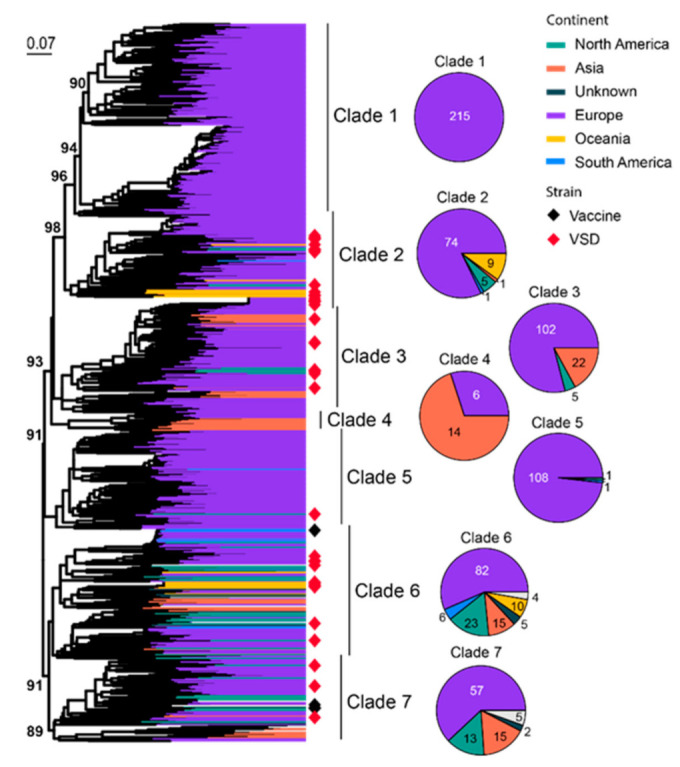
Diversity and phylogeographic structure of FCV based on VP1 gene. A maximum-likelihood tree reconstructed based on partial VP1 gene alignments (420 nt) is shown on the left, which includes 801 representative virus sequences obtained from the GenBank. The tree is mid-point rooted. Geographic locations (continent) for these virus strains are marked with different colors. Highly virulent strains and vaccine strains are labeled using red and black diamonds, respectively. The detailed geographic distributions of each subtype are shown as pie charts on the right of the phylogenetic tree.

**Figure 6 viruses-15-01338-f006:**
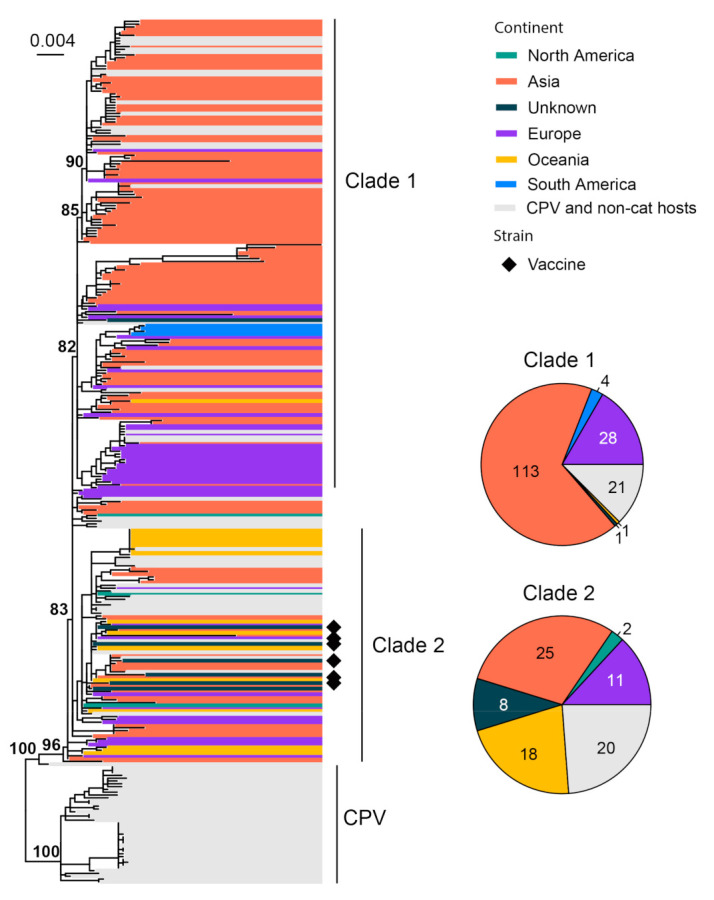
Diversity and phylogeographic structure of FPV based on VP2 gene. A maximum-likelihood tree reconstructed based on partial VP2 gene alignments (1755 nt) is shown on the left, which includes 252 virus sequences of FPV obtained from the GenBank. The tree is mid-point rooted. Geographic locations (continent) for these virus strains are marked with different colors. Vaccine strains are labeled using black diamonds. The detailed geographic distributions of each subtype are shown as pie charts on the right of the phylogenetic tree.

**Figure 7 viruses-15-01338-f007:**
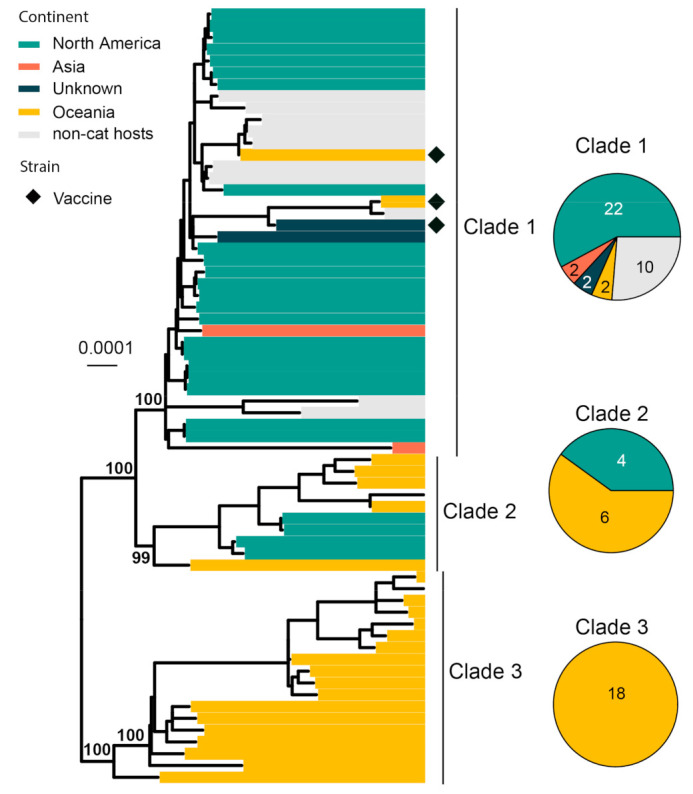
Diversity and phylogeographic structure of FHV-1 based on complete genome alignment. A maximum-likelihood tree reconstructed based on complete genome alignments (134,883 nt) is shown on the left, which includes 66 virus sequences obtained from the GenBank. The tree is mid-point rooted. Geographic locations (continent) for these virus strains are marked with different colors. Vaccine strains are labeled using black diamonds. The detailed geographic distributions of each subtype are shown as pie charts on the right of the phylogenetic tree.

**Figure 8 viruses-15-01338-f008:**
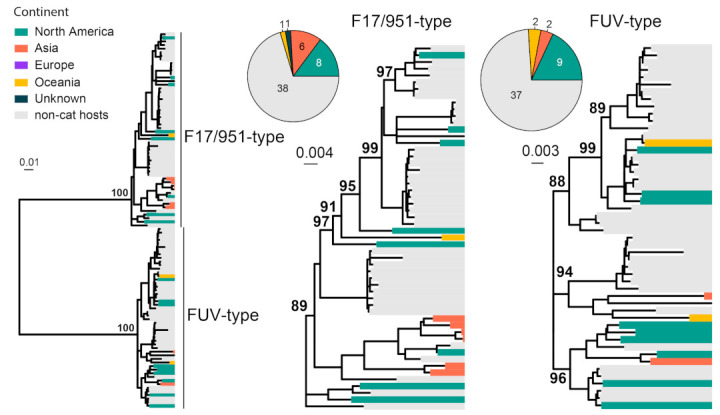
Diversity and phylogeographic structure of FFV based on env gene. A maximum-likelihood tree reconstructed based on env gene alignments (2949 nt) is shown on the left, which includes 104 virus sequences obtained from the GenBank. The tree is mid-point rooted. Geographic locations (continent) for these virus strains are marked with different colors. For each serotype, the geographic distributions are shown as a pie chart and placed left next to the corresponding subtree.

**Figure 9 viruses-15-01338-f009:**
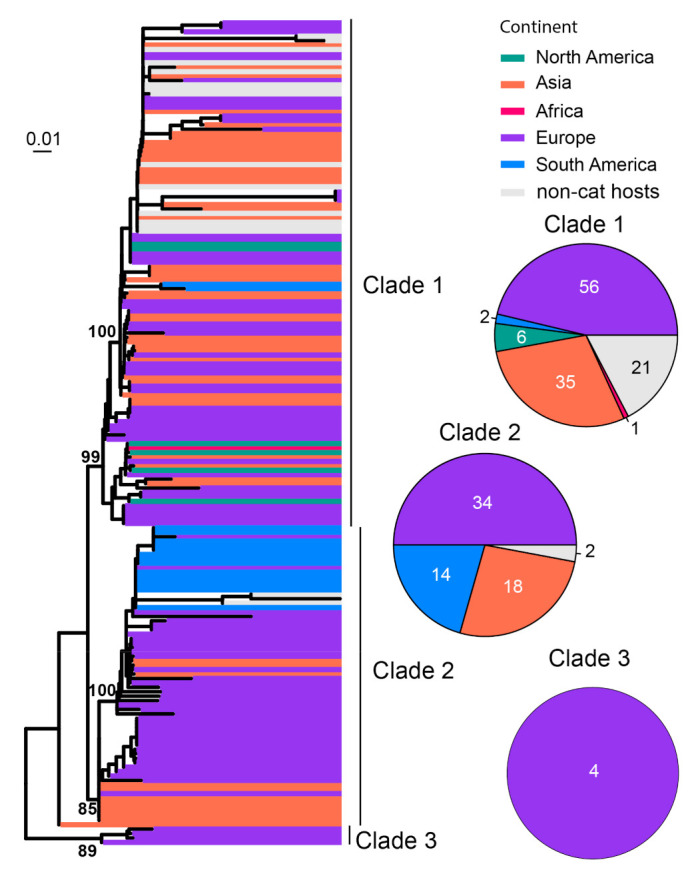
Diversity and phylogeographic structure of FeMV based on L protein gene. A maximum-likelihood tree reconstructed based on complete L protein gene alignment (6618 nt) is shown on the left, which includes 186 virus sequences obtained from the GenBank. The tree is mid-point rooted. Geographic locations (continent) for these virus strains are marked with different colors. The detailed geographic distributions of each subtype are shown as pie charts on the right of the phylogenetic tree.

**Figure 10 viruses-15-01338-f010:**
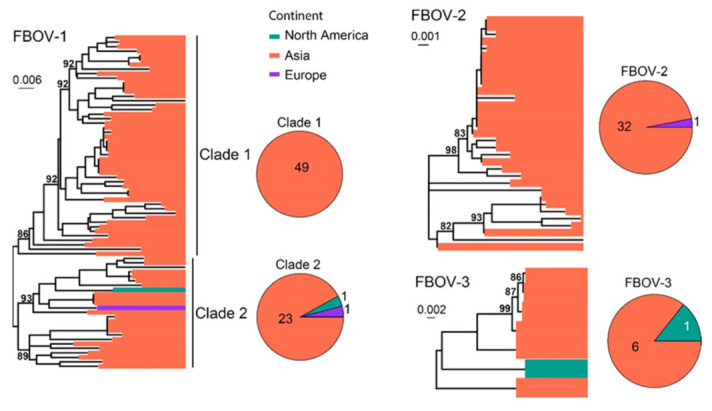
Diversity and phylogeographic structure of FBoVs based on NS1 gene. Three maximum-likelihood trees reconstructed based on complete NS1 gene alignments (2415 nt, 2394 nt and 2391 nt) are shown, which include 74, 33 and 7 virus sequences obtained from the GenBank, respectively. The trees are mid-point rooted. Geographic locations (continent) for these virus strains are marked with different colors. The detailed geographic distributions of each subtype are shown as pie charts on the right of the phylogenetic trees.

**Figure 11 viruses-15-01338-f011:**
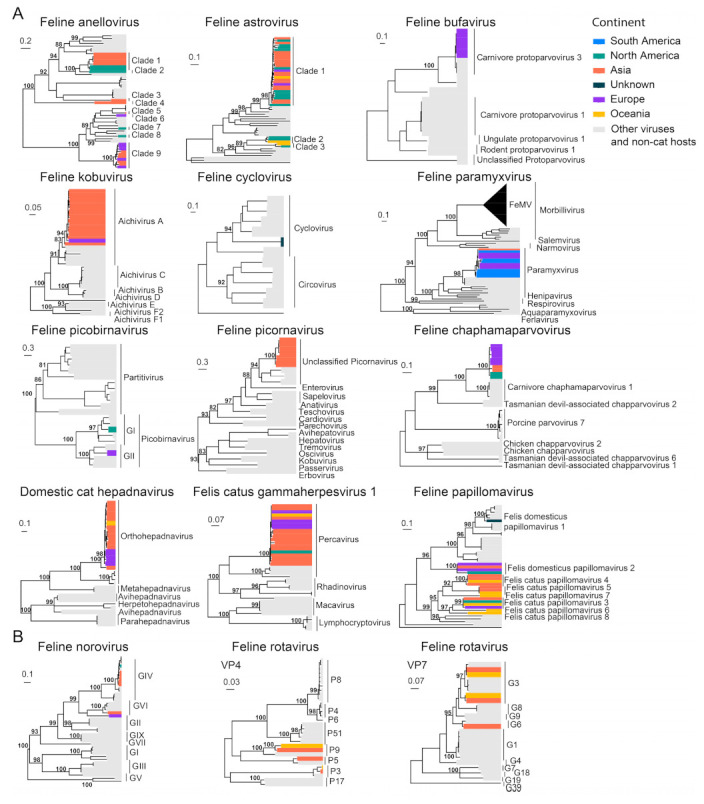
Diversity and phylogeographic structure of other feline viruses based on their marker genes/proteins. (**A**) Interspecific phylogenetic trees and (**B**) intraspecific phylogenetics were reconstructed for less characterized cat virus species. The trees are mid-point rooted. The virus species’ name is labeled on the top of each tree. Geographic locations (continent) for these virus strains are marked with different colors.

**Figure 12 viruses-15-01338-f012:**
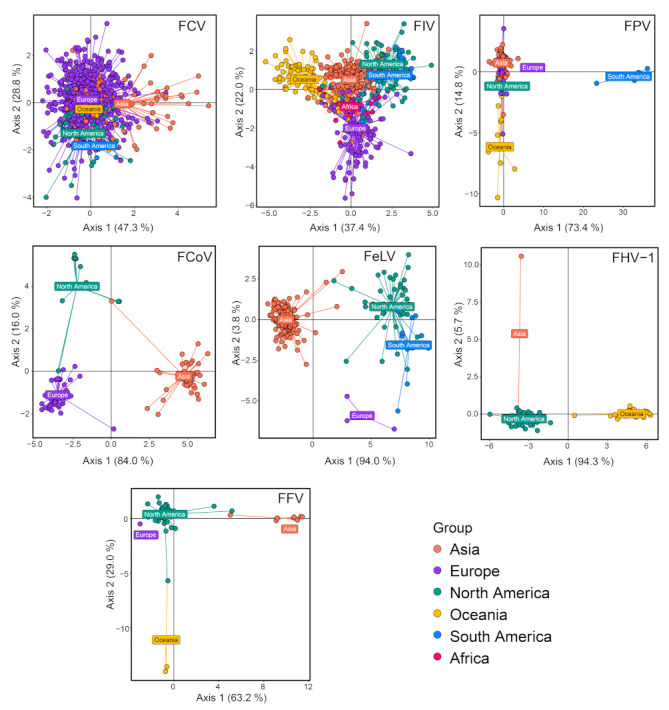
Discriminant analysis of principal components (DAPC) scatter plot of main feline viruses, including FCV, FIV, FPV, FCoV, FeLV, FHV-1 and FFV. Different colors indicate virus strains of different geographic regions. The *y*-axis and *x*-axis represent two principal discriminant components.

**Figure 13 viruses-15-01338-f013:**
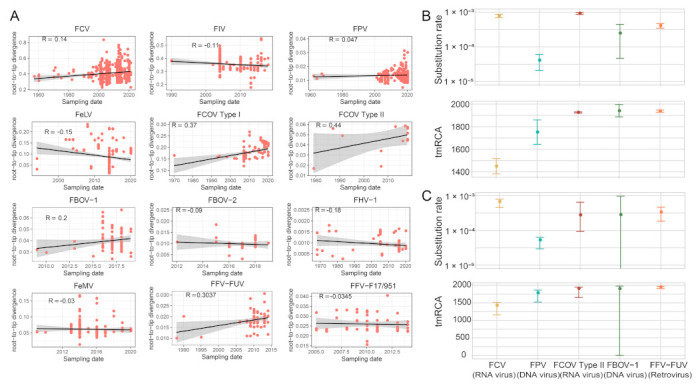
Evolutionary analysis. (**A**) Regression of root-to-tip genetic distance against the year of sampling for several feline viruses. Black indicates a linear regression line. (**B**) Estimates of evolutionary parameters (substitution rate and tMRCA) for feline viruses via TreeTime software (version 0.8.6). (**C**) Estimates of evolutionary parameters (substitution rate and tMRCA) for feline viruses via LSD software (version 0.3beta).

**Figure 14 viruses-15-01338-f014:**
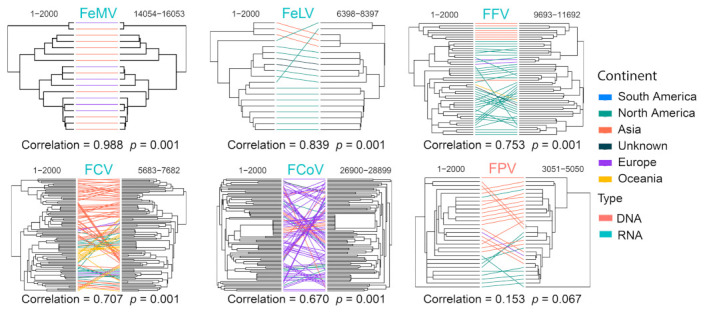
Recombination analyses of representative cat virus species. For each species, phylogenies based on the 5′-end and 3′-end of genomes are reconstructed and compared for potential signs of phylogenetic incongruence. The incongruence is measured by Pearson’s correlation coefficient shown on the bottom of the trees, which reflects the recombination frequency for this specific virus.

**Table 1 viruses-15-01338-t001:** The degree of geographic structure evaluated using Wilks’ lambda value.

Virus	Wilks’ Lambda	*p*-Value
FCV	0.63635	<0.001
FIV	0.06931	<0.001
FPV	0.00274	<0.001
FCoV	0.01490	<0.001
FeLV	0.02394	<0.001
FHV-1	0.02718	<0.001
FFV	0.01085	<0.001

## Data Availability

All alignments and phylogenetic trees in Newick format were included in the Appendix A “Alignments and trees.zip”.

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
