# Peer review of "Genetic Diversity and Evolution of Viruses Infecting Felis catus: A Global Perspective"

_viruses, 2023, doi:10.3390/v15061338_

Round 1

Reviewer 1 Report

This study presented an overview of the temporal-spatial distribution and evolutionary characterization of cat viruses using sequences from public databases. It provides some insights into the evolutionary and epidemiological feature of cat viruses and may also hep prevention and control of the cat virus. The analysis was conducted reasonably and the writing of the manuscript is clear. The reviewer has some minor concerns about the study.

1 It is suggested to add a discussion about the common features of cat viruses since the study analyzed the virus individually and nearly all results were obtained for some virus species.

2 The study analyzed the evolutionary features of different viral species based on marker genes of the virus. It is suggested to add more explanation about how to select the marker gene.

3 It is suggested to add an section of Conclusion which summaried the main findings of the study. Similarly, more results are suggested to be described in the Abstract.

4 The version of software tools used should be specified.

5 In Section 3.5, which correlation coefficient was used? It is not suitable to say correlation < 0.8.

6 There were some grammar errors such as following that the it was soon identified in Japan, 2.4 Analyse of phylogeographic structures

Author Response

Comment 1: It is suggested to add a discussion about the common features of cat viruses since the study analyzed the virus individually and nearly all results were obtained for some virus species.

Response 1: We agree with the reviewer and add the following “general discussion” to the context, to read,

 “Generally, a larger diversity and wider distribution is described for majority of viral species. Nevertheless, the diversity depicted here were by no means all-inclusive for these viruses. It is because the global sampling was largely uneven, with more diversity revealed for developed countries than developing ones. Indeed, the amounts of sequences collected from African countries were substantially smaller than Asian (mainly China and Japan), European, and North Americas. Furthermore, viruses with low pathogenicity or less clinical impact, namely, TTFV, FFV, FePV, FePBV, FeCyCV and FePaV, contained fewer sequences and poorer geographical representations, most likely due to lack of surveillance work. The global prevalence of these viruses is expected to be much higher given that many causes subclinical or latent infections in cats.”

Comment 2 The study analyzed the evolutionary features of different viral species based on marker genes of the virus. It is suggested to add more explanation about how to select the marker gene.

Response 2: The sequences were selected based on the number of sequences available, sequence divergence level, and the representativeness of diversity and geographic distribution. Corresponding description has been added to Methodology section of the manuscript.

Comment 3 It is suggested to add a section of “Conclusion” which summarized the main findings of the study. Similarly, more results are suggested to be described in the Abstract.

Response 3: We thank the reviewer for the advice, and a “Conclusion” section has been added to the manuscript, to read, “In summary, our study performed an all-inclusive sequence collection from the GenBank and a comprehensive phylodynamic analyses for 25 feline virus species. Our results revealed, for the first time, a global diversity much larger than previously de-picted, based on which we further characterized the geographic expansion patterns, temporal dynamics and recombination frequencies of these viruses. Importantly, the representative sequence data set obtained here forms knowledge basis for cat virus di-versity, evolution and epidemiology, and they could be used as reference data sets or diversity backbones for future survallance work.”

Furthermore, more results have been added in the “Abstract”, to read “While respiratory pathogen like feline calicivirus showed some degree of geographical panmixes, the other viral species are more geographically defined. Furthermore, recombination rates were much higher in feline parvovirus, feline coronavirus, feline calicivirus and feline foamy virus than the other feline virus species.”

Comment 4 The version of software tools used should be specified.

Response 4: We agree with the reviewer and the version of software tools has now been added.

Comment 5 In Section 3.5, which correlation coefficient was used? It is not suitable to say “correlation < 0.8”.

Response 5: We apologized for the confusion. It should be “Pearson’s correlation coefficient” and corresponding phrases have been corrected in the revised manuscript.

Comment 6 There were some grammar errors such as “following that the it was soon identified in Japan”, ”2.4 Analyse of phylogeographic structures”

Response 6: We apologized for these mistakes. The grammar errors have now been corrected.

Reviewer 2 Report

The study by Le et al describes the vast diversity of viruses found in cats. The authors use publicly available sequence data including the associated meta data and performed phylogenetic analyses to assess their evolutionary history and geographic distribution. The study investigates a lot of data and describes the genetic diversity of 25 virus species. The authors describe the different clades and genotypes and how they are distributed globally, the authors have also investigated temporal structure and evolutionary rate and finally recombination. This is the first study summarizing the genetic diversity of viruses in cats. The authors find that some viruses have a seemingly broader geographical distribution than others and present the differences in evolutionary rate and recombination for different virus species. Overall, the study is well present and very interesting.

 However, it’s a very long paper with a lot of data and different viruses included and I believe it could benefit with some structure and more detail in the figures to help with the understanding of the different viruses.

Main comments:

1.     The authors did not specify if the data used came for ‘targeted’ sequencing or metagenomic sequencing. The authors find vast differences in geographic distribution and point out potential sampling bias – if this is due to mainly targeted sequencing of pathogenetic viruses it should be noted in the manuscript.  

2.     How were clades defined? Sometimes the authors mentioned that clades or genotypes were defined as previously published, other time there is no such mention. It should be noted if there is no known topological separation. Particularly, the trees in figure 5,6 and 9 the clade assignment was less clear according to the topology.

3.     The study needs a concluding paragraph.

Minor comments

Figure 1. Great figure but I have some suggestion that I think will improve it even more.

A.    Add virus abbreviation as it makes it easier to compare to the text.

B.    I don’t know what the circle legends refers to. Is it number of sequences total per country? If so, I think it would be easier to just add the number next to the pie chart. For example North America n=13.

Figure 11. For some for the trees I would suggest reducing the number of ‘grey’ branches as it looks like there are no cat viruses in the tree. For example, Feline cyclovirus and bufavirus. I think this would help with visualisation.

Figure 12. Difficult to read the labels in the boxes. Maybe make the text white?

Figure 13. Figure 13BC needs thicker lines. Would suggest adding the gene and genome type to explain the differences in rates.

Figure 14. add genome type.

Line 9: The pathogen diversity in cats was always there, what we know about it has changed. I suggest rephrasing.

 ‘Cats harbor many important viral pathogens whose known diversity has increased thanks to increasingly popular molecular sequencing techniques.’

Line 24ff. I suggest rephrasing as it is a very long sentence with.

Since the discovery of first feline virus—feline parvovirus - in 1928 [1], more 24 than 23 feline virus species have been identified, among which six have been studied 25 extensively because of known disease association and/or worldwide prevalence. These 26 viruses include feline immunodeficiency virus (FIV), feline coronavirus (FCoV), feline leukemia 27 virus (FeLV), feline calicivirus (FCV), feline parvovirus (FPV) and feline? alphaherpesvirus 28 1 (FHV-1).’

Line 34.  …few unique diseases, such as acute mye-33 loid leukemia (AML), multicentric lymphoma, aplastic anemia, and other immunodeficiencies.

Line 35ff. Rephrase.

 In addition to these well-known pathogens, a number of other feline viruses were identified early, but their disease associations remained unclear.

Lines 51 and 52: should be cats instead of cat.

Line 56: Rephrase

… our knowledge on the geographic range of cat viruses has also expanded, thanks .

Lines 58: Rephrase.

…their geographic range have been shown to range from the place of discovery to across the globe.

Lines 60 ff. Rephrase

For example, FeMV was first discovered in Hong Kong in 2012 [16], and using PCR primers designed based on early discovered genomes it was identified also in Japan, the United Kingdom, Brazil, Turkey and Italy [36–43]. Similarly, FeChPV was first discovered in 2019 in a multi-facility feline shelter in Canada [5] before its discovery expanded to Turkey [44], Italy [25] and China [45,46].

There are many grammatical errors that need to be address but more importantly the authors refer to their study as ‘increasing the diversity’. I suspect this got lost in translation, as we don’t know much about the virus diversity in cats due to limited sampling. I think what the authors mean is that we increase the knowledge on the diversity and geographic spread. See some examples below.

Author Response

Main comments:

Comment 1.     The authors did not specify if the data used came for ‘targeted’ sequencing or metagenomic sequencing. The authors find vast differences in geographic distribution and point out potential sampling bias – if this is due to mainly targeted sequencing of pathogenetic viruses it should be noted in the manuscript.  

Response 1: We apologized for the confusion. The data were downloaded from the GenBank and therefore they should come from both ‘targeted’ sequencing and metagenomic sequencing. Nevertheless, since most of the sequencing was generated using PCR and sanger sequencing, it is reasonable to assume that majority of the sequences obtained here were from “targeted sequencing”. We have revised the corresponding text as follows:

“…contained fewer sequences and poorer geographical representations, most likely due to lack of surveillance from “targeted sequencing” …”

Comment 2.     How were clades defined? Sometimes the authors mentioned that clades or genotypes were defined as previously published, other time there is no such mention. It should be noted if there is no known topological separation. Particularly, the trees in figure 5,6 and 9 the clade assignment was less clear according to the topology.

Response 2: Despite of differences in diversity and sequence numbers, majority of the clades were derived from previously described groups and were well-supported. However, we agree with the reviewer that some of the clades were defined despite of low support value. It is because the lack of well-supported internal node to define such clade. The lack of support has been mentioned in the context, to read:

“… Within FPV, the diversity can be divided, with low confidence, into 2 major clades and a number of small transmission chains close to common ancestor of all FPVs (Figure 6) …”

Comment 3.     The study needs a concluding paragraph.

Response 3: We agree with the reviewer, and a “Conclusion” section has been added to the manuscript, to read, “In summary, our study performed an all-inclusive sequence collection from the GenBank and a comprehensive phylodynamic analyses for 25 feline virus species. Our results revealed, for the first time, a global diversity much larger than previously depicted, based on which we further characterized the geographic expansion patterns, temporal dynamics and recombination frequencies of these viruses. Importantly, the representative sequence data set obtained here forms knowledge basis for cat virus diversity, evolution and epidemiology, and they could be used as reference data sets or diversity backbones for future surveillance work.”

Minor comments

Comment 4. Figure 1. Great figure but I have some suggestion that I think will improve it even more.

  1. Add virus abbreviation as it makes it easier to compare to the text.
  2. I don’t know what the circle legends refers to. Is it number of sequences total per country? If so, I think it would be easier to just add the number next to the pie chart. For example North America n=13.

Response 4:

  1. Revised as suggested.
  2. The circle legends refer to the number of sequences. To make it easier to understand, we now add the numbers next to the pie chart instead of using the circle legends.

Comment 5. Figure 11. For some for the trees I would suggest reducing the number of ‘grey’ branches as it looks like there are no cat viruses in the tree. For example, Feline cyclovirus and bufavirus. I think this would help with visualisation.

Response 5: Revised as suggested.

Comment 6. Figure 12. Difficult to read the labels in the boxes. Maybe make the text white?

Response 6: Revised as suggested.

Comment 7. Figure 13. Figure 13BC needs thicker lines. Would suggest adding the gene and genome type to explain the differences in rates.

Response 7: Revised as suggested.

Comment 8. Figure 14. add genome type.

Response 8: Genome type has been added.

Comment 9. Line 9: The pathogen diversity in cats was always there, what we know about it has changed. I suggest rephrasing.

 ‘Cats harbor many important viral pathogens whose known diversity has increased thanks to increasingly popular molecular sequencing techniques.’

Response 9: Revised as suggested.

Comment 10. Line 24ff. I suggest rephrasing as it is a very long sentence with.

Since the discovery of first feline virus—feline parvovirus - in 1928 [1], more than 23 feline virus species have been identified, among which six have been studied extensively because of known disease association and/or worldwide prevalence. These viruses include feline immunodeficiency virus (FIV), feline coronavirus (FCoV), feline leukemia virus (FeLV), feline calicivirus (FCV), feline parvovirus (FPV) and feline alphaherpesvirus 1 (FHV-1).

Response 10: Revised as suggested.

Comment 11. Line 34.  …few unique diseases, such as acute mye-33 loid leukemia (AML), multicentric lymphoma, aplastic anemia, and other immunodeficiencies.

Response 11: Revised as suggested.

Comment 12. Line 35ff. Rephrase.

 In addition to these well-known pathogens, a number of other feline viruses were identified early, but their disease associations remained unclear.

Response 12: Revised as suggested.

Comment 13. Lines 51 and 52: should be cats instead of cat.

Response 13: Revised as suggested.

Comment 14. Line 56: Rephrase

… our knowledge on the geographic range of cat viruses has also expanded, thanks .

Response 14: Revised as suggested.

Comment 15. Lines 58: Rephrase.

…their geographic range have been shown to range from the place of discovery to across the globe.

Response 15: Revised as suggested.

Comment 16. Lines 60 ff. Rephrase

For example, FeMV was first discovered in Hong Kong in 2012 [16], and using PCR primers designed based on early discovered genomes it was identified also in Japan, the United Kingdom, Brazil, Turkey and Italy [36–43]. Similarly, FeChPV was first discovered in 2019 in a multi-facility feline shelter in Canada [5] before its discovery expanded to Turkey [44], Italy [25] and China [45,46].

Response 16: Revised as suggested.

Comment 17. Comments on the Quality of English Language

There are many grammatical errors that need to be address but more importantly the authors refer to their study as ‘increasing the diversity’. I suspect this got lost in translation, as we don’t know much about the virus diversity in cats due to limited sampling. I think what the authors mean is that we increase the knowledge on the diversity and geographic spread. See some examples below.

Response 17: We have thoroughly checked the main text, and the language has been improved.